# Assessment of Diet Quality in Chilean Urban Population through the Alternate Healthy Eating Index 2010: A Cross-Sectional Study

**DOI:** 10.3390/nu11040891

**Published:** 2019-04-20

**Authors:** Victoria Pinto, Leslie Landaeta-Díaz, Oscar Castillo, Luis Villarroel, Attilio Rigotti, Guadalupe Echeverría

**Affiliations:** 1Centro de Nutrición Molecular y Enfermedades Crónicas, Escuela de Medicina, Pontificia Universidad Católica de Chile, Santiago 8330033, Chile; vspinto@uc.cl (V.P.); arigotti@med.puc.cl (A.R.); 2Universidad Autónoma de Chile, Santiago 7500912, Chile; leslie.landaeta@uautonoma.cl; 3Escuela de Nutrición, Universidad Finis Terrae, Santiago 7501015, Chile; ocastillo@uft.cl; 4Departamento de Salud Pública, Escuela de Medicina, Pontificia Universidad Católica de Chile, Santiago 8330077, Chile; lv@med.puc.cl; 5Departamento de Nutrición, Diabetes y Metabolismo, Escuela de Medicina, Pontificia Universidad Católica de Chile, Santiago 8330077, Chile

**Keywords:** Chile, healthy diet, AHEI-2010

## Abstract

Most worldwide causes of disease and death are strongly associated with dietary factors and the application of eating indexes has proved to be a useful tool to determine diet quality in populations. The aim of this study was to evaluate the diet quality in Chile through the application of the Alternate Healthy Eating Index 2010 (AHEI-2010). A representative sample (*n* = 879) of Chilean urban population aged 15–65 years old from the Latin American Study of Nutrition and Health (*Estudio Latinoamericano de Nutrición y Salud*; ELANS) was used. Dietary intake data were obtained through two 24-hour food recalls and one beverage frequency questionnaire, which were used to calculate AHEI-2010 and its association with sociodemographic and anthropometric variables. In this Chilean sample, the AHEI-2010 score was 43.7 ± 7.8 points (mean ± SD). Trans fats and sodium intake were the highest scoring AHEI-2010 components whereas sugar-sweetened beverages and whole grains had the lowest score. Women, older subjects, and individuals in medium-high socioeconomic levels had significantly higher mean AHEI-2010 scores. No association was found between AHEI-2010 and body mass index (BMI), or nutritional status. Conclusions: Diet quality in the Chilean urban population aged 15–65 years old is far from optimal. Thus, there is room for significant improvement of diet quality in Chile through design and implementation of public health policies, particularly in high-risk groups for chronic diseases.

## 1. Introduction

Dietary factors are strongly associated with chronic noncommunicable diseases (NCDs), including cardiovascular diseases (CVD), type 2 diabetes, and certain types of cancer, and they rank among the top contributors to the burden of disease and death in developed and developing countries [1]. One approach to determine adherence to current nutritional recommendations is the application of dietary scores that evaluate overall food intake patterns. The Department of Nutrition at the Harvard School of Public Health developed in 2002 the Alternate Healthy Eating Index (AHEI) [2], which was updated in 2010 (AHEI-2010) [3]. This index was created with the purpose of improving the Healthy Eating Index 2005 (HEI-2005), developed by the United States Department of Agriculture (USDA) [4], and looking for a tool with better predictive value of the relationship between measurement of diet quality and risk of chronic diseases, mainly cardiovascular disease [3]. The AHEI-2010 is an 11-dimension dietary quality index that contains a profile of components derived from updated scientific evidence and dietary recommendations [3]. This new version is strongly associated with a lower risk of coronary heart disease and type 2 diabetes compared to HEI-2005 [3]. It has also been inversely related to other cardiovascular diseases, cancer [3], chronic obstructive pulmonary disease [5], and even depression [6]. This index was created based on a detailed food frequency questionnaire; nonetheless, it has also been calculated using data from a 24-h food recall [7,8].

Since 1960, there has been a lack of information about eating habits and food intake patterns in the overall Chilean population as well as in those groups with the greatest risk of diet-associated chronic diseases [9]. In 2010, a National Survey of Food Consumption was performed in a representative Chilean population sample to identify eating patterns and to provide background information for design of local dietary guidelines and public policymaking [10]. Data obtained from this survey was used to calculate a Spanish adaptation of HEI, which established that only 5.3% of the studied Chilean sample matched the healthy eating category [10]. Furthermore, a healthy diet index was also created based on very limited food intake information collected during the 2009–2010 National Health Survey. This index demonstrated a low diet quality in Chileans ≥18 years old and was inversely associated with the presence of metabolic syndrome [11]. In both studies, a higher diet quality was identified in women, older subjects, and high socioeconomic level [10,11]. 

Despite its extensive validation and significant disease predictive value, AHEI-2010 has not been applied to measure diet quality either in Chile or in other Latin American countries. This world region is undergoing an alarming epidemiological transition from nutritional deficiencies to overweight and obesity as well as other eating pattern-related chronic diseases [12,13]. Thus, the use of well-validated dietary indexes is critical to evaluate time trends as well as inter-country and interregional comparisons of diet quality.

The purpose of this study was to evaluate the diet quality in Chile through the application of the AHEI-2010 using the database of the Chilean sub-sample recruited for the Latin American Nutrition and Health Study (“Estudio Latinoamericano de Nutrición y Salud”; ELANS) [14].

## 2. Materials and Methods 

### 2.1. Study Sample

ELANS is a household-based multinational cross-sectional survey carried out in 2014–2015 and aimed at evaluating food and nutrient intake in nationally representative samples of urban populations (aged 15 to 65 years old) from Argentina, Brazil, Chile, Colombia, Costa Rica, Ecuador, Peru, and Venezuela [14]. All participants signed assent and/or informed consent. As exclusion criteria were considered individuals below 15 or over 65 years old, pregnant and lactating women, individuals with major physical or mental impairments that affected food intake and physical activity, adolescents without assent and consent of a parent or legal guardian, individuals living in any residential setting other than a household, and individuals unable to read [14]. The survey included two household visits for each participant in which sociodemographic information, anthropometric measurements, diet, and physical activity evaluations were collected. Further details of the study design and logistics are published elsewhere [14]. The overall ELANS protocol was approved by Western Institutional Review Board (#20140605) and is registered at clinicaltrials.gov website (#NCT02226627). In addition, the Pontificia Universidad Católica de Chile Research Ethics Committee locally approved the ELANS protocol applied in Chile. 

With regard to this study, we used the Chilean subsample of ELANS corresponding to 879 subjects with complete and valid data (Figure 1). 

### 2.2. Dietary Assessment

Dietary intake data were obtained using two 24-h food recalls (24-HR) performed on two non-consecutive days within one week, considering weekend days proportionally to weekdays. The second visit also considered application of a beverage frequency questionnaire, which included 10 beverage categories (water, flavored water, soft drinks, fruit drinks, sport drinks, energy drinks, tea and coffee drinks, other non-alcoholic drinks, and alcoholic drinks). For each beverage, participants answered whether they consumed the specific category of beverage, the frequency of intake (daily, weekly, monthly), and how often they drink the beverage during the selected unit (1–10 occasions).

Each recall was administered by trained interviewers, provided with standardized neutral probing questions according to the Multiple Pass Method [15] and accompanied with a graphic portions booklet, to improve precision of the information obtained. The interviewers were trained to collect all measurements by certified nutritionists/dietitians who simultaneously worked as supervisors of the fieldwork. Estimated energy, macronutrient, and micronutrient intakes were obtained using the Nutrition Data System for Research software, version 2013 (NDS-R, Minnesota University, MN, USA) [16]. The Multiple Source Method (MSM) was used for adjustment to estimate usual dietary intake from the two 24-HR. The MSM method is a web-based statistical modeling, available at MSM website (https://msm.dife.de/tps/en), which was developed by researchers at the European Prospective Investigation into Cancer and Nutrition (EPIC) [17].

### 2.3. AHEI-2010 Application

AHEI-2010 total score was calculated as the sum of individuals’ scores of its 11 components. Nine out of 11 component scores of AHEI-2010 were calculated with the information obtained from the two 24-HR after processing by the NDS-R nutritional software and MSM statistical adjustment. Intake of vegetables (excluding potatoes and juices), fruits (excluding juices), whole grains (including brown rice, popcorn, and any grain food with a carbohydrate-to-fiber ratio ≤10:1), nuts and legumes and red and processed meat were used as servings/day. Intake of nutrients such as trans fat and polyunsaturated fatty acids (PUFAs) was used as percentage of total energy/day, and long-chain (ω-3) fats (eicosapentaenoic and docosahexaenoic acids, EPA + DHA) and sodium as mg/day. Information for sugar-sweetened beverages and fruit juices and alcohol in servings/day was obtained from the beverage frequency questionnaire. Nutrient contributions from dietary supplements were not included [3]. All AHEI-2010 components were scored from 0 to 10 points as shown in Table 2. For fruits, vegetables, whole grains, nuts and legumes, long-chain (ω-3) fats, and PUFAs, an increasing score corresponded to higher intake. For trans fat, sugar-sweetened beverages and fruit juices, red and/or processed meat and sodium, a higher score corresponded to lower intake. For alcohol, the highest score was assigned to moderate intake (0.5–1.5 drinks/day in women and 0.5–2 drinks/day in men), and the lowest score to heavy alcohol consumers. Nondrinkers received a score of 2.5 points. The total AHEI-2010 score ranged from 0 (nonadherence) to 110 (optimal adherence to a healthy diet) [3,7]. 

### 2.4. Anthropometric Assessment 

Anthropometric measurements of body weight and height and were obtained according to standardized procedures [14]. Body weight was measured with a calibrated electronic scale up to 200 kg with an accuracy of 0.1 kg. Height was measured with a portable stadiometer up to 205 cm with an accuracy of 0.1 cm. Each measurement was repeated twice to ensure accuracy, and the average was used for further analyses. Categorization of nutritional status by BMI in adolescents (15–19 years old) was based on the gender-specific BMI-for-age cut-off points from the World Health Organization (WHO) [18]. For adults (older than 19 years), BMI was categorized as underweight (<18.5 kg/m^2^), normal weight (18.5–24.9 kg/m^2^), overweight (25.0–29.9 kg/m^2^), and obesity (≥30.0 kg/m^2^).

### 2.5. Statistical Analysis

All analyses incorporated sampling weights to allow inferences applicable to the overall Chilean urban population older than 15 years. Kolmogorov-Smirnov test was used to determine if AHEI-2010 score had a normal distribution. Parametric Student’s t-test was used to compare specific component and overall AHEI-2010 scores by sex. One-way ANOVA with Bonferroni’s multiple comparisons test was used for categorical variables (age group, socioeconomic levels (SEL), and nutritional status). In addition, Pearson correlation coefficient was calculated to assess the association between the AHEI-2010 score and numerical anthropometric variable (BMI).

We used terciles to divide the sample in three groups (i.e., low, intermediate, and high scoring) based on AHEI-2010 calculations. To compare these groups according to participants’ sociodemographic, anthropometric, and nutritional intake (total energy and macronutrients intake) characteristics, we used Chi-square test whereas one-way ANOVA was applied for continuous variables (total energy and macronutrient intake). 

A univariate linear regression analysis was used to examine adjusted associations between independent variables (total energy intake (continuous, kcal/day), sex (categorical: men, women), age group (categorical: 15–19, 20–34, 35–49, 50–65 years), SEL (categorical: low, medium, high), and nutritional status (categorical: underweight, normal, overweight, obesity)) with AHEI-2010 score. 

All analyses were conducted with SPSS^®^ Statistics, version 24 (IBM Corporation, Armonk, NY, USA). All *p* values were two-tailed with α = 0.05.

## 3. Results

879 Chilean subjects (50.8% women) living in urban areas and aged 15 to 65 years old completed the survey. The mean ± SD of age was 37.5 ± 13.9 years old and body mass index (BMI) was 28.4 ± 5.6 kg/m^2^. Distribution of socioeconomic levels (SEL) was representative of the overall country (Table 1). 

### 3.1. Overall AHEI-2010 and AHEI-2010 Component Scores

The AHEI-2010 score in the urban Chilean population had a normal distribution (Kolmogorov Smirnov test, *p* = 0.200), with a mean score of 43.7 ± 7.8 points, ranging from 20.1 (least healthy diet) to 69.4 (healthiest intake) points. Table 2 shows AHEI-2010 components ordered from highest to lowest mean scores separated by those with positive versus negative relation between intake and score (except the alcohol intake component). Trans fat, sodium, red/processed meat, and PUFA were the components with higher scores, all of them with mean scores over 50% of maximum score (>5 points). The alcohol component also averaged more than 5 points, but only in men. On the other hand, fruits, sugar-sweetened beverages and fruit juices, and whole grains were the components with the lowest scores. Appendix A visually depicts which components have higher and lower mean scores. 

### 3.2. Overall AHEI-2010 and AHEI-2010 Component Scores by Sociodemographic Characteristics and Nutritional Status

Significant differences in mean AHEI-2010 scores were observed by sex, age group and SEL (Figure 2). Women exhibited higher AHEI-2010 scores than men (45.8 versus 41.6 points, *p* < 0.001). AHEI-2010 also increased with age (*p* for trend <0.001), from 41.0 points in subjects 15 to 19 years old, to 45.6 points in subjects older than 50 years of age. Participants with low SEL had significantly lower mean AHEI-2010 scores than participants in medium and high SEL (*p* < 0.001). With regard to nutritional status, we also found significant differences in AHEI-2010 scores (*p* for trend <0.05). In particular, obese subjects have higher mean AHEI-2010 scores than normal weight subjects (Figure 2). 

When analyzing AHEI-2010 components by sex, women exhibited higher scores in fruits, whole grains, red/processed meats, PUFA and sodium, and lower scores in nuts and legumes and alcohol compared to men. Subjects in older groups also showed significant higher scores in all AHEI-2010 components, except for whole grains and EPA+DHA, in which no differences were found by age groups. We also detected significant differences in SEL, where subjects with medium and high —compared to low— SEL had higher scores in vegetables, fruits, whole grains, red/processed meats, trans fat, sodium, and sugar-sweetened beverages and fruit juices. 

### 3.3. Sociodemographic, Anthropometrics, and Nutritional Intake According to Tercile Distribution of AHEI-2010 Score

Characteristics of the subjects and description of AHEI-2010 score based on tercile distribution of this diet quality index are shown in Table 3. The high AHEI-2010 score group compared with the low AHEI-2010 score group had a significantly higher percentage of women (64.6% versus 33.6%, respectively), older subjects (30.6% versus 17.1% of individuals older than 50 years old), participants of medium-high SEL (65.4% versus 40.7%, respectively), and obese subjects (39.9% versus 29.4%, respectively). In addition, subjects in the high score group reported significantly lower total energy and relative carbohydrate (as % of total energy) intake as well as higher relative protein and fat intake (as % of total energy) than lower AHEI-2010 score groups. 

### 3.4. Correlation between AHEI-2010 Score andBMI 

A statistically significant, though weak, direct correlation (*p* = 0.003, *r* = 0.101) was found between AHEI-2010 score and BMI. 

### 3.5. Univariate Linear Regression Model

Differences in AHEI-2010 scores by sex, age group, and SEL remained statistically significant when these variables were integrated, together with energy intake and nutritional status, in a multivariable linear regression model (Table 4). Nevertheless, the association between nutritional status and AHEI-2010 score no longer remained significant (*p* = 0.168). Keeping constant the remaining independent variables included in the model, women scored on average 2.1 points higher in AHEI-2010 than men, subjects between 15–19 years old had 3.9 points less than those 50–65 years old, and subjects in low SEL scored 3.9 points lower than those at high SEL.

## 4. Discussion

Despite its extensive validation and significant disease predictive value in North America [3,5,7,19], the current study is the very first report on the application of AHEI-2010 as a measure of diet quality in Latin America using a representative sample of Chilean urban population. 

The mean AHEI-2010 score of Chilean sample was 43.7 of a total of 110 points. Even though AHEI-2010 does not have cut-off points, the mean value obtained is far from optimal. Other dietary indexes have been applied in Chilean samples: a Spanish adaptation of the original HEI for the National Survey of Food Consumption sample database [10] and a healthy diet index created ad hoc with food intake information of the 2009–2010 National Health Survey [11]. Both studies concluded that diet quality of the Chilean population was deficient (low consumption of fish, legumes, vegetables and fruits, and a high intake of sugars and refined grains) and needed improvement. Nonetheless, the results of these studies are not comparable with ours based on the differences in the methodology as well as dietary indexes applied.

The United States AHEI-2010 mean score is higher than that obtained in our study (46.8 versus 43.7, respectively), based in the 2009–2010 US National Health and Nutrition Examination Survey (NHANES) [7]. This difference may be explained by the use of industrial production estimates of trans fat from the FDA in the US sample rather than measures of actual intake [7]. When comparing both scores with exclusion of the trans fats component, Chile exhibits a slightly better diet quality than the USA (37.9 points in our Chilean sample versus 37.1 points in the USA NHANES sample). 

Regardless of these methodological differences, the highest score component of AHEI-2010 in Chile also was trans fats (a higher score meaning lower intake). Trans fats intake are associated with higher risk of coronary heart disease (CHD) [20] and diabetes [21]. Since 2009, trans fats content of industrial foods has been limited in Chile to ≤2% of total fat content [22]. Furthermore, WHO plans to eliminate industrially produced trans fatty acids from global food supply by 2023 [23]. In 2004, Denmark became the first country in the world to regulate the content of artificial trans fat in certain ingredients for food production [24], which nearly eliminated it from the Danish food supply. This food policy has been followed by a decrease in mortality rates due to CVD [24]. This measure is feasible to be applied in our country to improve diet quality. 

The second highest score component was sodium (a higher score meaning lower intake). High sodium intake has been associated with higher blood pressure [25], and salt-preserved foods are associated with greater risk of stomach cancer [26], CVD [27], and total mortality [28]. Furthermore, sodium-reduced diets significantly lowered blood pressure [29] and CVD risk in clinical trials [30]. In our study, sodium score was obtained only from its content in foods (natural and processed), but added salt was not assessed. Even though sodium is the second highest score component, it does not mean that its consumption in Chile is optimal: the mean for sodium was 5.8 out of 10, which is actually not high at all. If we consider that approximately 80% of sodium is provided by foods sources [31,32], the estimated mean total daily consumption of salt raises to 7.6 g. This amount is above the WHO recommendation of 5.0 g/day [33]. More recently, the 2016–2017 Chilean National Health Survey (NHS) showed an average salt consumption of 9.4 g/day, with the 98% of the population above the WHO recommendation [34]. Negative front-of-pack labeling warning about sodium content in processed foods was implemented in Chile in mid-2016 [22]. This measure has been already associated with decreased sales of these unhealthy products [35]; however, it is too early to correlate it with changes in clinical outcomes. 

Although red and processed meats had the third highest AHEI-2010 score (a higher score meaning a lower intake), its consumption is still high considering the current dietary recommendations, with an average intake of 0.6 servings/day. This level of intake is consistent with the information provided by the National Food Consumption Survey in 2014, which reported a median consumption of 83 g/day of red meats and 26 g/day of processed meats [10]. High consumption of red and processed meats has been associated with several chronic disease conditions —especially cardiovascular diseases and cancer [36,37].

On the other hand, the two components with the lowest scores were sugar-sweetened beverages (SSB) and fruit juices as well as whole grains. With regard to the first food item, Chile had the fastest global absolute growth on sales of SSB between 2009 and 2014 [38]. Intake of SSB, including soda and fruit drinks, is associated with increased risk of weight gain and obesity [39], CVD [40], and type 2 diabetes [41]. This component also includes intake of fruit juices, which shows positive association with diabetes risk [42] and lack of beneficial effects on CVD [43]. In the current study, we also included in this component homemade beverages prepared with sugar, such as tea and coffee, because its high consumption in our country (added sugars represents 22.0% of total carbohydrates intake [44] and 22.8% of added sugar intake comes from homemade beverages (unpublished data) in Chilean ELANS sample). In our country, an additional 5% tax was implemented on non-alcoholic beverages with more than 6.25 g of sugar per 100 ml, which has been enforced since 2014 [45]. However, national and international evidence suggests that a tax increase <10% may be insufficient to generate a significant impact on health outcomes [46,47]. Additional actions were taken in 2016 including reduction of SSB availability in schools, restrictions in marketing of sugary foods to children, and negative front-of-pack labeling. This latter measure has also impacted sales of SSB [35].

Greater consumption of whole grains is well associated with lower risk of CVD [48], diabetes [49], and colorectal cancer [26]. However, this component had the lowest score both in Chile and in the USA [7]. No whole grains intake analysis had been previously reported in Chilean population, but it has been recently reported by the ELANS group [50] that whole grains products consumption is low in our country as well as in other countries from Latin America, representing less than 1% of total energy intake. This low consumption may be related to its low palatability (astringent texture and low softness and humidity), unawareness of its health benefits, and higher prices compared with refined grains/cereals [51].

The AHEI-2010 defines the highest score of alcohol intake to moderate consumption due to its association with lower risk of CHD [52], diabetes [53], and all-cause and CVD mortality [54]. Even though men had a higher score than women in this component (5.2 versus 3.8 points) and mean consumption of alcohol was less than 0.5 drinks/day in both sexes, other studies have reported that Chile has the first place in alcohol intake in Latin America [55]. Indeed, this consumption is characterized by a binge-drinking pattern, concentrated in only 1.6 days/week on average [55]. Thus, the intake observed in our study probably was not as the moderate daily consumption that has been associated with beneficial clinical outcomes. 

Additionally, fruits and vegetables had <50% of the maximal score, reflecting a very low consumption of these healthy foods. Similar results were observed in the 2016–2017 Chilean NHS with only 15% of the population reporting an intake of at least five servings/day of fruits and vegetables [56]. Vegetables and fruits consumption have been associated with lower risk of CVD [57,58] and some cancers [59,60]. It is important to consider that AHEI-2010 takes as highest score a consumption of four and five servings/day of fruits and vegetables, respectively, whereas the Chilean government promotes a less stringent consumption of five servings/day of combined vegetable and fruit intake. 

On the other hand, the 2016–2017 NHS also reported that 24.4% of the Chilean population consumed two servings/week of legumes and only 9.2% at least two servings/week of fish [34], which is consistent with low consumption of legumes and EPA+DHA observed in our study. The low consumption of legumes may due to the long preparation time required before intake and the popular belief that legume use is related to poverty [61]. With regard to fish, high prices are the main reason why consumers limit its intake in our country [62]. 

With regard to sociodemographic variables analyzed in our study, AHEI-2010 score had a sociodemographic distribution similar to other indexes previously applied in Chile [10,11,63]. As expected, women had better diet quality than men as well as older subjects and medium-high SEL groups. In Chile, the prevalence of chronic diseases considerably increases in older subjects [56] and may determine a greater concern when choosing their foods. Socioeconomic level was also inversely associated with dietary quality and there are some potential explanations for this disparity. First, healthy foods generally cost more than unhealthy foods [64] and limited access to healthy foods also affects choices in low SEL. Nutrition education and knowledge, which is strongly related to socioeconomic level, is also likely to play an important role in adoption of healthy dietary habits [7].

With regard to nutritional status and AHEI-2010 score, no relation was found after adjustments by sex, age group, SEL, and total energy intake. In addition, weak positive correlations were found between BMI and AHEI-2010 score. These findings are not consistent with some studies that have demonstrated a greater diet quality in subjects with lower BMI [18,65,66]. Nonetheless, this trend was quite modest and may be influenced by potential confounders because nutritional status was not adjusted by other sociodemographic variables such as sex, age, or SEL. In addition, we cannot rule out a social desirability bias when participants answered the 24-HR recalls or the beverage frequency questionnaire [14]. 

Despite the differences found between sociodemographic subgroups (sex, age and SEL), each of them presented an average score of AHEI-2010 less than 50, indicating that the Chilean diet is far from being healthy, representing an important risk factor in the current epidemic of obesity and chronic diseases affecting the country.

Most of the studies applying AHEI-2010 have reported its longitudinal association with clinical outcomes by terciles or quintiles of score. These studies have shown that the lowest score groups had worst clinical outcomes with more risk of CVD, type 2 diabetes, cancer and total mortality [2,3,18,65,66]. We postulate that our lowest AHEI-2010 score group, composed mainly by men, younger subjects and low SEL, may also have an increased risk of CVD and mortality. However, actual prospective studies are required to test this hypothesis. Meanwhile, it is important to implement public interventions aimed at increasing diet quality in this risk group. A few changes, such as a rise in consumption of nuts and/or legumes from no servings to one serving/day and a reduction in intake of red/processed meats from 1.5 servings/day to lower or zero consumption, will result in a 20 points improvement in the overall score. Assuming a causal relationship, a person who increases the AHEI-2010 score in 20 points over a 12-year period may reduce his or her risk of death by nearly 20% in the subsequent 12 years, based on estimations from previous studies [19]. 

The present study has limitations. First, ELANS has a cross-sectional design, which limits causal and time-based inferences. Second, misreporting of food intake is one of the main sources of error when using dietary assessment instruments based on self-report (e.g., 24-HR recalls). Under reporting occurs in most adult populations, especially in women and in those persons with a higher BMI [14]. In addition, we cannot rule out a social desirability bias when participants answered the 24-HR recalls or the beverage frequency questionnaire. Despite of these limitations, this report contributes to a limited body of literature about diet quality in 15 to 65-year-old Chileans living in urban locations. Future prospective cohort studies in more representative Chilean populations should determine if diet quality measured by AHEI-2010 is associated with various clinical outcomes such as CVD and mortality. 

## 5. Overall Conclusions and Potential Implications

Diet quality of the Chilean urban population aged 15–65 years old is far from optimal; thus, there is room for significant improvement in this lifestyle component. As mentioned, it is necessary to assess the impact of some public policies already implemented in our country. Additional measures are also required to impact economically disadvantaged and high-risk groups. If so, advances in overall diet quality may lead to noncommunicable chronic disease prevention in our population.

## Figures and Tables

**Figure 1 nutrients-11-00891-f001:**
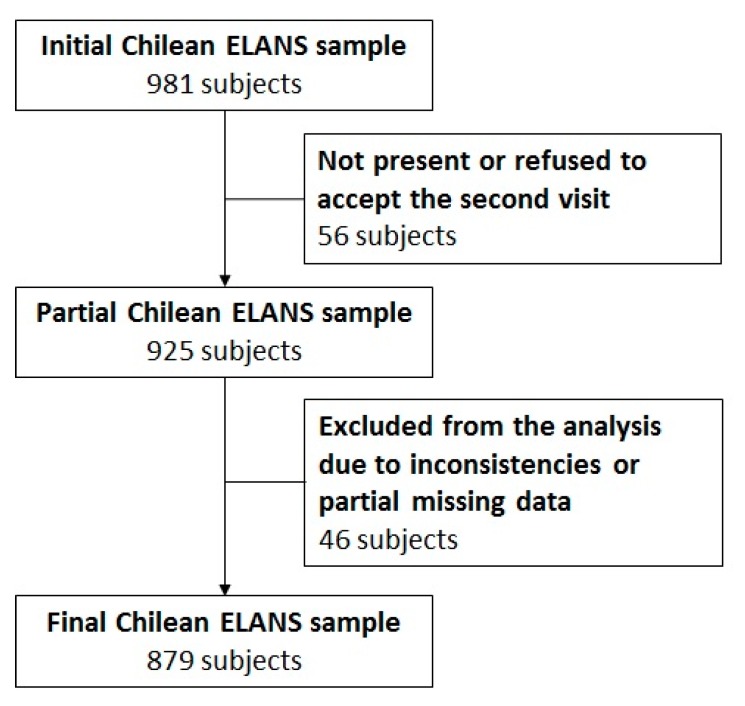
Flow diagram of the Chilean sample enrolled in the Latin American Study of Nutrition and Health (“Estudio Latinoamericano de Nutrición y Salud”; ELANS).

**Figure 2 nutrients-11-00891-f002:**
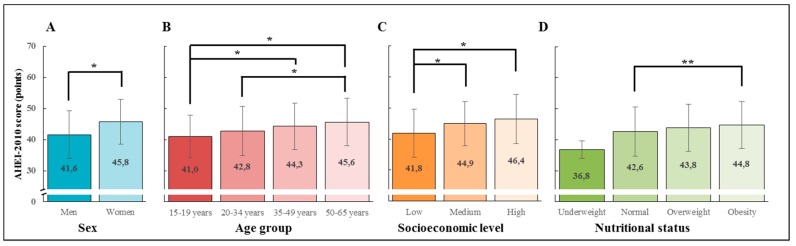
Alternate Healthy Eating Index 2010 scores according to sex, age group, socioeconomic level and nutritional status in Chilean urban population enrolled in ELANS 2014–2015. Mean ± SD scores for AHEI-2010 are shown by sex (**A**), age groups (**B**), socioeconomic level (**C**), and nutritional status (**D**). Differences between groups were considered statistically significant if *p* < 0.05 using parametric Student’s *t*-test (for sex) or one-way ANOVA with Bonferroni’s multiple comparisons test (for age groups, socioeconomic level, and nutritional status). * *p* ≤ 0.001 ** *p* < 0.05.

**Table 1 nutrients-11-00891-t001:** Characteristics of Chilean participants enrolled in ELANS 2014–2015.

Sociodemographic Characteristics and Nutritional Status	Frequency *n* (%) ^†^
Total sample	879 (100)
Sex	
Women	446 (50.8)
Men	433 (49.2)
Age group	
15–19 years	91 (10.3)
20–35 years	307 (35.0)
35–50 years	263 (29.9)
50–65 years	218 (24.8)
Socioeconomic level	
Low	400 (45.4)
Medium	338 (38.5)
High	141 (16.1)
Nutritional status	
Underweight	3 (0.3)
Normal	266 (30.3)
Overweight	315 (35.8)
Obesity	295 (33.6)

† Numbers of subjects presented as raw data while frequencies are adjusted by sampling weights.

**Table 2 nutrients-11-00891-t002:** Scoring method and mean intakes and scores of each Alternate Healthy Eating Index 2010 component in Chilean urban population enrolled in ELANS 2014–2015.

Component	Criteria for Minimum Score (0)	Criteria for Maximum Score (10)	Mean Intake	Mean Score
Positive relation between score and intake				
PUFA, % of total energy/day	≤2	≥10	6.4 ± 1.2	5.5 ± 1.5
Vegetables, servings/day	0	≥5	2.4 ± 0.8	4.7 ± 1.6
Long-chain (ω-3) fats (EPA + DHA), mg/day	0	≥250	67.3 ± 53.0	2.6 ± 1.6
Nuts and legumes, servings/day	0	≥1	0.2 ± 0.3	2.0 ± 2.3
Fruits, servings/day	0	≥4	0.7 ± 0.6	1.8 ± 1.6
Whole grains, servings/day				
Women	0	≥5	0.4 ± 0.6	0.7 ± 1.3
Men	0	≥6	0.3 ± 0.7	0.5 ± 1.2
Total sample				0.6 ± 1.2
Negative relation between score and intake				
Trans fat, % of total energy/day	≥4	≤0.5	1.0 ± 0.2	8.7 ± 0.5
Sodium, mg/day	Highest decile ^†^	Lowest decile ^†^	2438.2 ± 786.3	5.8 ± 3.2
Red/processed meat, servings/day	≥1.5	0	0.6 ± 0.3	5.8 ± 1.7
SSB and fruit juice, servings/day	≥1	0	2.8 ± 2.3	1.6 ± 3.1
Nonlinear relation between score and intake				
Alcohol, drinks/day				
Women	≥2.5	0.5–5	0.1 ± 0.2	3.8 ± 2.1
Men	≥3.5	0.5–2.0	0.3 ± 0.7	5.2 ± 2.8
Total sample				4.5 ± 2.6
**Total**	**0**	**110**	-	**43.7 ± 7.8**

Components are ordered from highest to lowest mean Alternate Healthy Eating Index 2010 (AHEI-2010) scores separated by those with positive versus negative relation between score and intake. Bold number represent the total AHEI-2010 score. †Values in lowest decile were 1,654 mg/day and in the highest decile were 3445 mg/day. PUFA: polyunsaturated fatty acids; EPA + DHA: eicosapentaenoic and docosahexaenoic acids; SSB: sugar sweetened beverages.

**Table 3 nutrients-11-00891-t003:** Description of Alternate Healthy Eating Index 2010 score categorized in by terciles according to sociodemographic characteristics, nutritional status and dietary intake (energy and macronutrients) characteristics in Chilean urban population enrolled in ELANS 2014–2015.

Sociodemographic Characteristics, Nutritional Status and Dietary Intake	Low AHEI-2010	Intermediate AHEI-2010	High AHEI-2010	*p*
Descriptive statistics				
*N*	293	293	294	
Mean ± SD	35.5 ± 0.2	43.4 ± 0.1	52.4 ± 0.3	<0.001
Median (range)	36.1	43.4	51.7	<0.001
	(20.1–40.2)	(40.2–46.7)	(46.7–69.4)	
Sociodemographic characteristics				
Sex (%)				
Women	33.6	54.1	64.6	<0.001
Men	66.4	45.9	35.4	
Age group (%)				
15–19 years	15.4	8.6	6.8	<0.001
20–34 years	42.3	33.2	29.3	
35–49 years	25.3	31.2	33.3	
50–65 years	17.1	27.1	30.6	
Socioeconomic level (%)				
Low	59.2	42.5	34.6	<0.001
Medium	30.8	42.9	41.7	
High	9.9	14.6	23.7	
Nutritional status				
Nutritional status (%)				
Underweight	1.0	0.0	0.0	0.003
Normal	36.2	30.5	24.2	
Overweight	33.4	38.4	35.8	
Obesity	29.4	31.2	39.9	
Energy and macronutrients				
Total energy intake (kcal/day)	1968.1	1638.3	1500.7	<0.001
Carbohydrate (% of total energy)	54.0	55.2	53.7	0.026
Protein (% of total energy)	15.4	16.1	17.2	<0.001
Fat (% of total energy)	29.9	29.1	30.3	0.018

Chi-square test and one-way ANOVA (for waist circumference, energy and macronutrients) were applied to compare characteristics of 879 participants as a function of terciles of AHEI-2010 score. *p* < 0.05 were considered statistically significant.

**Table 4 nutrients-11-00891-t004:** Adjusted linear regression model of associations between Alternate Healthy Eating Index 2010 with sociodemographic characteristics and nutritional status in Chilean urban population enrolled in ELANS 2014–2015.

Sociodemographic Characteristics and Nutritional Status	*n*	β	95% Confidence Interval		*p*
			Lower Limit	Upper Limit	
Sex					<0.001
Women	426	2.15	1.13	3.17	<0.001
Men	412	0			
Age group					<0.001
15–19 years	92	−3.88	−5.68	-2.09	<0.001
20–34 years	298	−1.80	−3.07	0.53	0.005
35–49 years	247	−0.21	−1.48	1.07	0.752
50–65 years	201	0			
Socioeconomic level					<0.001
Low	366	−3.96	−5.27	-2.65	<0.001
Medium	325	−0.98	−2.31	0.34	0.146
High	147	0			
Nutritional status					0.168
Underweight	4	−5.68	−12.45	1.10	0.100
Normal	261	−1.08	2.30	0.14	0.083
Overweight	289	−0.52	−1.64	0.61	0.367
Obesity	284	0			

Univariate linear regression model for AHEI-2010 total score adjusted for total energy intake (kcal, continuous), sex (women, men (reference)), age group (15–19, 20–34, 35–49, 50–65 (reference) years-old), socioeconomic level (low, medium, high (reference)), and nutritional status (underweight, normal, overweight, obesity (reference)).

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
