# Peer review of "Assessment of Diet Quality in Chilean Urban Population through the Alternate Healthy Eating Index 2010: A Cross-Sectional Study"

_nutrients, 2019, doi:10.3390/nu11040891_

Round 1
Reviewer 1 Report
This manuscript was very well written and informative. Methodology and statistical analysis sections were easy to read and clearly written. The purpose of the study was to use the AHEI 2010 to describe the diet quality of a Chilean population. The main finding was that the overall AHEI score was lower than what is recommended and that various segments of the population for example, women, older subjects and those individuals at medium to higher socioeconomic levels scored higher than their counterparts. Overall AHEI scores were relatively low and the authors concluded that much research work is needed, primarily using prospective cohorts, to understand the long term implications of these findings.
I wondered about two questions as I read the paper:
How close is the current dietary profile of Chile in comparison to Western-style diet? The authors suggest that this might be so as they report that the people of Latin American are adapting eating behaviors that are related to chronic disease.
Since the AHEI-2010 was created based on dietary habits and relation to chronic disease risk of mainstream U.S. populations, are the components of the AHEI-2010 reflective of traditional dietary habits of the people of Chile? For example, nuts? Whole grains? Are the components sensitive to pick up traditional food practices that might be reflective of better diet quality?
Also, NDSR, did it provide sufficient food groups/food choices to pick up on the consumption of traditional foods/staples? Using 24 hour recalls is important to capture these differences in traditional versus mainstream foods, but sometimes there are limitations when this data is then analyzed using NDSR or similar tools. Did the authors encounter any issues? How was this streamlined? Any comments about this?
Under Dietary Assessment
Was dietary data explored for plausibility? If so, authors should add this information to this section as it is an important part of examining dietary data.
Under 3.5 Multivariable linear regression model
Line 217: Fix typo from "women scored in average..." to "women scored an average..."
Author Response
I wondered about two questions as I read the paper:
How close is the current dietary profile of Chile in comparison to Western-style diet? The authors suggest that this might be so as they report that the people of Latin American are adapting eating behaviors that are related to chronic disease.
Previous studies conducted in the Chilean population have shown that the Chilean diet is far from being considered healthy. Based on a healthy eating index adapted from Spain, the National Survey of Food Consumption (ENCA) conducted in 2010-2011 showed low consumption of fish, legumes, vegetables and fruits and a high intake of sugars and refined grains. As a consequence, 94.7% of the population requires changes in their diet towards a healthier pattern (1). This finding is cited in lines 253-260 in the new version of the article.
In addition, our own studies performed in more than 50,000 Chilean adults, in which diet quality was assess applying a Mediterranean dietary index, reported that only 9.5% of Chileans have a high adherence to the Mediterranean diet and the most common pattern includes high intake of sugar and sugary foods, fatty dairy, red and processed meats together with low consumption of lean and white meats, legumes, fish and seafood, and mono- and polyunsaturated fats (2). In this same sample of Chilean adults, we ha also reported that a lower diet quality (e.g., low adherence to a Mediterranean dietary pattern) was strongly associated with overweight and obesity as well as metabolic syndrome (3).
References:
(1) Universidad de Chile. Encuesta Nacional de Consumo Alimentario Informe Final. [Internet]. Santiago de Chile: Universidad de Chile, Departamento de Nutrición 2016 Feb 2 [cited 2018 May 20]. Available from http://www.minsal.cl/sites/default/files/ENCA-INFORME_FINAL.pdf/. Spanish.
(2) [Validation of self-applicable questionnaire for a Mediterranean dietary index in Chile]. Echeverría G, Urquiaga I, Concha MJ, Dussaillant C, Villarroel L, Velasco N, Leighton F, Rigotti A. Rev Med Chil. 2016 Dec;144(12):1531-1543. doi: 10.4067/S0034-98872016001200004
(3) Inverse Associations between a Locally Validated Mediterranean Diet Index, Overweight/Obesity, and Metabolic Syndrome in Chilean Adults. Echeverría G, McGee EE, Urquiaga I, Jiménez P, D'Acuña S, Villarroel L, Velasco N, Leighton F, Rigotti A. Nutrients. 2017 Aug 11;9(8). pii: E862. doi: 10.3390/nu9080862.
Since the AHEI-2010 was created based on dietary habits and relation to chronic disease risk of mainstream U.S. populations, are the components of the AHEI-2010 reflective of traditional dietary habits of the people of Chile? For example, nuts? Whole grains? Are the components sensitive to pick up traditional food practices that might be reflective of better diet quality?
Yes, components included in the AHEI-2010 are consistent with dietary habits of the Chilean population. Indeed, food items such as nuts, whole grains, fish, seafood, legumes, fruits and vegetables are commonly used in traditional Chilean culinary practices and available through the food market.
Also, NDSR, did it provide sufficient food groups/food choices to pick up on the consumption of traditional foods/staples? Using 24 hour recalls is important to capture these differences in traditional versus mainstream foods, but sometimes there are limitations when this data is then analyzed using NDSR or similar tools. Did the authors encounter any issues? How was this streamlined? Any comments about this?
ELANS included an a priori harmonization process that checked nutritional equivalency of food items reported by the study subjects in each country against the US version of foods available in the NDSR database (1). To ensure maximal similarity with Latin American foods, US foods were identified in this database through its general description (i.e., by name, type, and mode of preparation). If an item consumed in Latin America was not available in this software, a food with similar class definition, description, and nutrient content was considered. Finally, very specific regional/local foods and recipes as well as non equivalent commercial foods not present in the NDSR database were broken down into ingredients and entered into the software as user recipes.
Reference
(1) Kovalskys I, Fisberg M, Gómez G, Rigotti A, Cortés LY, Yépe, MC, Pareja RG, Herrera-Cuenca M, Zimberg IZ, Tucker KL, et al. Standardization of the food composition database used in the Latin American Nutrition and Health Study (ELANS). Nutrients 2015; 7: 7914-7924.
Under Dietary Assessment. Was dietary data explored for plausibility? If so, authors should add this information to this section as it is an important part of examining dietary data.
This is a very important comment that we will consider in future publications. In this paper, we have not analyzed the data for plausibility, which will require detailed analysis and consensus of the ELANS group in order to choose, calibrate, and validate the best method to address this issue However, we are aware that food intake underreporting is a weakness in this type of studies and thus may be affecting our findings on the application of the AHEI in Chile. A comment on this potential limitation was included in lines 377-381.
Nonetheless, only 6 (vegetables, fruits, nuts, legumes, EPA+DHA, whole grains, and red/processed meats) out of the 11 components of AHEI-2010 are the most susceptible food items to be underreported because their scores were calculated based on g or mg/day or servings/day. In contrast, alcohol and sugar-sweetened beverages scores were obtained from the food frequency questionnaire so they are less likely to be affected by underreport. The sodium score was adjusted based on the total sample salt intake, thus it was not expressed as absolute consumption, but relative to the overall study sample. Finally, PUFA and trans fat scores were calculated as % of total energy intake, so they were corrected for possible underreport.
Under 3.5 Multivariable linear regression model. Line 217: Fix typo from "women scored in average..." to "women scored an average..."
This typo error was fixed in the new version of the manuscript.

Reviewer 2 Report
This study aimed to evaluate the diet quality of a nationally representative sample of 15-65 year old Chileans living in urban locations using the AHEI-2010. The authors present an important need for the study, and the study findings are quite interesting and relevant. However, there are several areas in the manuscript that should be elaborated on and made clearer. I also found that the discussion could be more concise and focus better on interpreting what was actually found. My comments for specific sections of the manuscript can be found below:
Introduction
1. Reference 1 does not seem to support the opening statement that dietary factors are strongly associated with NCDs. Please provide a citation for studies that strongly associate dietary factors with CVD, type 2 diabetes, and certain types of cancer.
Methods
Study sample
1. The statement on lines 87-88 is not clear as written. Please rewrite to make clearer.
2. Although the ELANS study design is published elsewhere, it would be helpful to briefly provide more information about the survey. For example, because you excluded 56 subjects who were not present or refused to accept the second visit, it would be helpful for the reader to know that ELANS conducts two visits.
3. You have not yet described some key information that is included in Table 1 (e.g. how some of your variables used in Table 1 were measured, that you incorporated sampling weights). I suggest moving Table 1 and the description of characteristics of Chilean participants (lines 89 – 92) to the beginning of the results section.
4. Table 1: Nutritional status adds up to 880 (3 + 267 + 315 + 295). Please correct. Please write out ELANS and include dates in the Table 1 title. Please include mean BMI and waist circumference in the table.
Dietary assessment
5. One line 100, please include recalls to the statement “two 24-HR”.
6. Please cite the Multiple Pass Method. Was a companion booklet used to estimate portion sizes?
7. Please provide more detail about the beverage frequency questionnaire. What beverages were included, What was the time frame, etc.
8. What output did NDS-R provide that was able to be used to calculate AHEI-2010 scores?
9. Were the two 24-hr recalls averaged after being entered into NDS-R? Please elaborate.
10. Please briefly describe the Multiple Source Method and cite.
11. I am confused what “Details of dietary assessments are described in previous articles” refers to. What other relevant details are described? Please make this statement clearer.
12. How was the beverage frequency questionnaire incorporated into the analysis?
Anthropometric assessment
13. Please include equipment used for anthropometric assessment.
14. How was weight status determined for subjects >19 years old? Please explain and cite.
15. I am also confused by the statement “More details about anthropometric measurements are reported elsewhere”. Please be specific. What other relevant details are reported elsewhere?
AHEI-2010 application
16. I suggest reorganizing this section to come immediately after the Dietary assessment section.
17. Please reword lines 118-119. The sentence is not clear as written.
18. Please include in this section how components were scored (e.g. servings/day, mg/day, etc) and that criteria for scores was different for men and women. While this information is included in Table 2, it would also be helpful for the reader to have it included in this section.
19. Line 129: In parentheses, please include how moderate alcohol intake is defined.
20. Line 131 – 132: As I’ve commented in other sections, the statement “More details about each component of AHEI-2010 are described in previous articles” is confusing. Please explain exactly what details can be found in these articles that may be relevant to the reader.
21. Line 133: Please include “recalls” after 24-HR.
22. What do you mean by “both 24-HR were used to calculate scores”? Did you take the average? If you averaged the 24-hour recall data, please explain in the Dietary assessment section. Did you use the output from NDS-R to calculate AHEI-2010 scores? Were food groups used for certain components (i.e. vegetables, fruits, whole grains, etc)? Please make this clearer.
23. Please see my comment 7 above – more information is needed about the beverage frequency questionnaire.
24. Table 2: Please look over for consistency (e.g. shouldn’t whole grains should say “servings” not “serving”). Please include the survey used and include dates in the table title.
Statistical analysis
25. Why were age, sex, socioeconomic level, and nutritional status selected as characteristics to compare diet quality? Were they selected a priori based on the evidence in other Chilean studies of diet quality? Are there any other important characteristics that the ELANS survey includes that have been associated with diet quality in other populations and may be important to assess?
Additional comments about methods:
26. Please include information about other variables included in Table 1 (e.g. were sex and age self-reported? How was socioeconomic level determined?). You could merge this detail into the section with anthropometric assessment information and rename the section.
27. Similar to how you looked at both mean and categorical BMI, could you categorize waist circumference based on recommendations and examine the association with diet quality? Please cite the waist circumference recommendations you use for adults and adolescents and include in the section about anthropometric assessments.
28. When you say that terciles were divided based on AHEI-2010 calculations do you mean based on distribution of the scores? Or are there guidelines for low, intermediate, and high AHEI-2010 scores. Please explain.
29. Please briefly explain why you examined AHEI-2010 score both continuously and categorically.
30. Lines 149 – 151 are not clear as written. Do you mean that all the independent variables were entered simultaneously and adjusted for total energy intake? Why wasn’t waist circumference included in the regression? What were the reference groups for each independent variable in the regression?
Results
Section 3.1
31. Table 2: For alcohol and whole grains: Could you take the average of the mean scores for men and women and report in the table? Might this change how these components would be organized in the table if you are ordering from highest to lowest mean score?
32. Table 2: In my opinion, it would make more sense to organize the table by those components in which higher score indicates higher intake (fruits, vegetables, whole grains, nuts and legumes, long-chain (ω-3) fats and PUFA) followed by the components in which higher score indicates lower intake (trans fat, sugar-sweetened beverages and fruit juices, red and/or processed meat and sodium). Then, you could include a radar chart that visually depicts which components had higher and lower mean scores. Please consider.
Section 3.2
33. What were effect sizes for these significant differences? Please include how you determined effect sizes in the Statistical analysis section and then write what the effect sizes were in the Results.
34. Please refer to Figure 2 at the end of line 169.
35. Figure 2 provides a very nice visual description of the results.
Section 3.3
36. Lines 193 – 194: I see what you’re saying by stating that the high AHEI-2010 score group indicated a healthier diet, but this is misleading because out of the total score (110) none of these groups really displayed “healthier” diets. The mean scores for all three groups are really quite poor. Instead, please consider writing that subjects in the high AHEI-2010 score group had higher AHEI-2010 total scores than those in the low AHEI-2010 score group.
37. Please include in the Statistical analysis section that you assessed the association between total energy and macronutrients and AHEI-2010 tercile.
38. Table 3: please provide p-value for the mean score.
Section 3.4
39. Please reword lines 215 – 216: Consider: the association between nutritional status and AHEI-2010 score no longer remained significant.
40. Line 217: do you mean “on average” instead of “in average”?
Additional comments:
41. Consider moving results about basic sample characteristics to the beginning of the results section (see comment 3).
42. Please make sure that all figures and tables include the survey name and dates of the survey in the title.
Discussion
43. Please report an overall evaluation of the AHEI-2010 mean score of your sample in the first paragraph of your discussion (i.e. did mean score indicate suboptimal or optimal overall diet quality based on the total score out of 110?). What are implications for suboptimal overall diet quality and health outcomes?
44. Could you also describe how overall diet quality in your study compares to other studies that have assessed diet quality of Chileans using other diet quality indexes?
45. Lines 229 – 231: If this was something that you examined statistically, it should be written in the Statistical analysis section and reported in the results. Otherwise, please just state that the AHEI-2019 scores were lower in the Chilean sample compared to the U.S. and do not provide a p-value.
46. If you will refer to the United States as US, please write United States (US) in line 229.
47. Lines 232 – 234: comment 45 applies.
48. Lines 249 – 250: Even though sodium is the second highest score component, it does not mean that its consumption in Chile is optimal – please do not confuse having the second highest score with meaning the score was high. The mean for sodium was 5.8 out of 10, which is actually not high at all. Unless the AHEI-2010 provides standards for each component (i.e. what scores are considered “high” vs “poor”) please refrain from using this misleading wording. As you state in your discussion, the sodium intake in Chile is suboptimal, and your results support this statement.
49. Lines 282 – 285: Even though men had a high score in this component (5.2 points) – please reword this to a higher score than women, as a score of 5.2 out of 10 is not really “high” (see comment 47). In addition, consuming 0 drinks/day gives a score of 2.5. Many studies show that consuming no alcohol has health benefits, however, because the AHEI assesses alcohol intake based on moderate consuming, consuming 0 drinks/day has implications for the mean component score. This may be something you want to address in your discussion.
50. Lines 281 – 288: Are you implying that because you only had two 24-hour recalls you may not have captured binge-drinking behavior that has been reported in another Chilean study? I’m not sure you can make this implication based on your results. However, you could state something about limitations of 24-hour recalls in this section. Please rewrite lines 286 -288.
51. Even though the Chilean government provides less stringent fruits and vegetables guidelines, average fruits and vegetables intake in your study was 0.7 servings/day and 2.4 servings/day, respectively. Please elaborate on the potential implications of consuming inadequate fruits and vegetables and health, as you did with other components in previous sections.
52. While the reasoning behind why some of the scores may have scored as the highest or lowest components is interesting, I think this is misleading to readers. Besides trans fat, all of the averages were below 6 out of 10. Sodium and red/processed meat essentially had the same mean score, while the mean value for fruits and SSB were also quite similar. Instead of focusing on the two highest components and the two lowest components, please consider the implications for mean scores as most component scores would likely be considered inadequate. This is why I suggested ordering your table differently and using a radar chart to display results, so you are focused on mean scores rather than order of scores in the table (see comment 32).
53. Lines 299 – 301: if the association is not significant, please do not state that individuals with a lower SEL had a higher consumption of legumes.
54. What about a discussion about the red/processed meat component score?
55. Please cite this statement (lines 318-319): In addition, we cannot rule out a social desirability bias when participants answered the 24-HR recalls or the beverage frequency questionnaire.
56. Could you please write about the clinical significance of the significant associations you found since overall mean AHEI-2010 was <50 across sex, age, SEL, and nutritional status?
57. Please remove lines 323 – 324 (We postulate that our lowest AHEI-2010 score group, composed mainly by men, younger subjects and low SEL, may also have an increased risk of CVD and mortality) and make sure your discussion focuses on interpreting your results. This statement is very strong considering your study was cross-sectional and mean AHEI-2010 score was quite low in all terciles.
58. Please remove lines 329 – 331 (Assuming a causal relationship, a person who increases the AHEI-2010 score in 20 points over a 12-year period may reduce his or her risk of death by nearly 20% in the subsequent 12 years, based on estimations from previous studies). You could instead write “Previous studies suggest that improvements in AHEI-2010 score may reduce risk of mortality over time”.
59. Please refrain from stating that “this report is one of the best available” (lines 336). Instead, you could state that your findings contribute to a limited body of literature about diet quality in 15 – 65-year-old Chileans living in urban locations.
Conclusions
60. I find that the conclusions stray from the purpose and findings of this cross-sectional study and could be rewritten.
61. This statement on lines 344 – 346 is too strong based on your study design and findings: If so, advances in overall diet quality will lead to noncommunicable chronic disease prevention and increased life quality of our population.
Additional comments about manuscript:
62. I noticed several minor grammatical errors throughout. Please proofread and make sure tense is consistent throughout manuscript.
Author Response
Introduction
Reference 1 does not seem to support the opening statement that dietary factors are strongly associated with NCDs. Please provide a citation for studies that strongly associate dietary factors with CVD, type 2 diabetes, and certain types of cancer.
Reference 1 was eliminated from this statement and all reference numbers were corrected
Methods
Study sample
1. The statement on lines 87-88 is not clear as written. Please rewrite to make clearer.
This sentence was changed in the new version of the article (lines 90-91)
2. Although the ELANS study design is published elsewhere, it would be helpful to briefly provide more information about the survey. For example, because you excluded 56 subjects who were not present or refused to accept the second visit, it would be helpful for the reader to know that ELANS conducts two visits.
More information on ELANS study design and protocol was given in the new version of the article (lines 83-85)
3. You have not yet described some key information that is included in Table 1 (e.g. how some of your variables used in Table 1 were measured, that you incorporated sampling weights). I suggest moving Table 1 and the description of characteristics of Chilean participants (lines 89 – 92) to the beginning of the results section.
Table 1 was moved to the Results section of the manuscript (lines 164-169)
4. Table 1: Nutritional status adds up to 880 (3 + 267 + 315 + 295). Please correct.
This issue was corrected in the new version of the article
5. Please write out ELANS and include dates in the Table 1 title. Please include mean
BMI and waist circumference in the Table.
The title of this Table was corrected. BMI and means are described in the text (lines
165-166). Waist circumference information was removed from the article because no
significant difference was found and did not provide extra information beyond BMI
analysis.
Dietary assessment
6. One line 100, please include recalls to the statement “two 24-HR”.
24-HR is the abbreviation for “24-hour food recall” (line 96).
7. Please cite the Multiple Pass Method. Was a companion booklet used to estimate portion sizes?
Reference included [15]. We used a graphic portions booklet to help the interviewers to better estimate portion sizes (lines 104-105).
8. Please provide more detail about the beverage frequency questionnaire. What beverages were included, What was the time frame, etc.
More details of the beverage frequency questionnaire are included in the article (lines 98-102).
9. What output did NDS-R provide that was able to be used to calculate AHEI-2010 scores?
This issue is now addressed in the article (lines 114-131).
10. Were the two 24-hr recalls averaged after being entered into NDS-R? Please elaborate.
An explanation on this comment was added to the article (lines 114-131).
11. Please briefly describe the Multiple Source Method and cite.
As requested, this method is now described and cited (lines 110-112, reference [17]).
12. I am confused what “Details of dietary assessments are described in previous articles” refers to. What other relevant details are described? Please make this statement clearer.
More information was added to describe dietary assessments (lines 94-112).
13. How was the beverage frequency questionnaire incorporated into the analysis?
The beverage frequency questionnaire was used as now described in lines 96-102.
Anthropometric assessment
14. Please include equipment used for anthropometric assessment.
Details of the equipment are included in the article (lines 135-137).
15. How was weight status determined for subjects >19 years old? Please explain and cite.
Further details are described in the article (lines 140-142)
16. I am also confused by the statement “More details about anthropometric measurements are reported elsewhere”. Please be specific. What other relevant details are reported elsewhere?
Further details are included in lines 133-142.
AHEI-2010 application
17. I suggest reorganizing this section to come immediately after the Dietary assessment section.
This section was reorganized as suggested by the reviewer.
18. Please reword lines 118-119. The sentence is not clear as written.
This sentence was rewritten for better comprehension (lines 114-131).
19. Please include in this section how components were scored (e.g. servings/day, mg/day, etc) and that criteria for scores was different for men and women. While this information is included in Table 2, it would also be helpful for the reader to have it included in this section.
Information is included in Table 2 and the scoring is also described in the article (lines 114-131).
20. Line 129: In parentheses, please include how moderate alcohol intake is defined.
Definition of moderate alcohol intake is included in the article (line 128).
21. Line 131 – 132: As I’ve commented in other sections, the statement “More details about each component of AHEI-2010 are described in previous articles” is confusing. Please explain exactly what details can be found in these articles that may be relevant to the reader.
Additional details on AHEI-2010 are described in lines 114-131.
22. Line 133: Please include “recalls” after 24-HR.
As said before, 24-HR means “24-hour dietary recall”
23. What do you mean by “both 24-HR were used to calculate scores”? Did you take the average? If you averaged the 24-hour recall data, please explain in the Dietary assessment section. Did you use the output from NDS-R to calculate AHEI-2010 scores? Were food groups used for certain components (i.e. vegetables, fruits, whole grains, etc)? Please make this clearer.
Further details are provided in the new version of the article (lines 114-131).
24. Please see my comment 7 above – more information is needed about the beverage frequency questionnaire.
This issue was fixed in lines 96-102.
25. Table 2: Please look over for consistency (e.g. shouldn’t whole grains should say “servings” not “serving”). Please include the survey used and include dates in the table title.
This comment was addressed in Table 2.
Statistical analysis
26. Why were age, sex, socioeconomic level, and nutritional status selected as characteristics to compare diet quality? Were they selected a priori based on the evidence in other Chilean studies of diet quality? Are there any other important characteristics that the ELANS survey includes that have been associated with diet quality in other populations and may be important to assess?
All these sociodemographic characteristics have been associated with diet intake in previous food surveys in Chile, such as National Health Survey 2009-2010 and 2016-2017 as well as the National Food Consumption Survey 2014)
Additional comments about methods:
27. Please include information about other variables included in Table 1 (e.g. were sex and age self-reported? How was socioeconomic level determined?). You could merge this detail into the section with anthropometric assessment information and rename the section.
Sex and age were self-reported. Socioeconomic level was evaluated by questionnaires using a format that was country-dependent and based on the legislative requirements or established local standard layouts. Socioeconomic data was divided into three strata (high, medium and low) based on the national indexes used in each country. Reference: Asociacion Investigadores de Mercado. Grupos Socioeconómicos Chile. Chile: Asociacion Investigadores de Mercado; 2012.
28. Similar to how you looked at both mean and categorical BMI, could you categorize waist circumference based on recommendations and examine the association with diet quality? Please cite the waist circumference recommendations you use for adults and adolescents and include in the section about anthropometric assessments.
Waist circumference information was removed from the article because no significant difference was found and did not provide extra information beyond BMI analysis.
29. When you say that terciles were divided based on AHEI-2010 calculations do you mean based on distribution of the scores? Or are there guidelines for low, intermediate, and high AHEI-2010 scores. Please explain.
There are no defined cut-off points for the AHEI-2010. We used terciles based on AHEI-2010 score distribution.
30. Please briefly explain why you examined AHEI-2010 score both continuously and categorically.
AHEI-2010 does not define categories, but in previous publications (cited in the article) the authors have divided their samples in terciles in order to analyze differences between groups. We used continuous AHEI-2010 to describe associations and differences between sociodemographic variables.
31. Lines 149 – 151 are not clear as written. Do you mean that all the independent variables were entered simultaneously and adjusted for total energy intake? Why wasn’t waist circumference included in the regression? What were the reference groups for each independent variable in the regression?
This section has been rewritten in the article (lines 157-160).
Results
Section 3.1
32. Table 2: For alcohol and whole grains: Could you take the average of the mean scores for men and women and report in the table? Might this change how these components would be organized in the table if you are ordering from highest to lowest mean score?
If we considered the total average for whole grains and alcohol the order of the AHEI-2010 components did not change. As suggested, we also included these averages in Table 2.
33. Table 2: In my opinion, it would make more sense to organize the table by those components in which higher score indicates higher intake (fruits, vegetables, whole grains, nuts and legumes, long-chain (ω-3) fats and PUFA) followed by the components in which higher score indicates lower intake (trans fat, sugar-sweetened beverages and fruit juices, red and/or processed meat and sodium). Then, you could include a radar chart that visually depicts which components had higher and lower mean scores. Please consider.
As suggested by the reviewer, we included a radar chart as a Supplementary Figure 1 (lines 582-588).
Section 3.2
34. What were effect sizes for these significant differences? Please include how you determined effect sizes in the Statistical analysis section and then write what the effect sizes were in the Results.
The calculation of effect sizes is relevant when you want to calculate a sample size or when comparing a treated group versus a control group in an intervention study. In this study, we analyzed the differences in AHEI-2010 score according to sociodemographic variables and statistical tests used were ANOVA or T-Student for independent samples as described in the Methods section. No other statistical test is required to assess the size of the differences between groups.
35. Please refer to Figure 2 at the end of line 169.
See lines 189-190.
36. Figure 2 provides a very nice visual description of the results.
Thanks for this positive comment.
Section 3.3
37. Lines 193 – 194: I see what you’re saying by stating that the high AHEI-2010 score group indicated a healthier diet, but this is misleading because out of the total score (110) none of these groups really displayed “healthier” diets. The mean scores for all three groups are really quite poor. Instead, please consider writing that subjects in the high AHEI-2010 score group had higher AHEI-2010 total scores than those in the low AHEI-2010 score group.
This sentence was fixed as indicated by the reviewer,(lines 214-215).
38. Please include in the Statistical analysis section that you assessed the association between total energy and macronutrients and AHEI-2010 tercile.
Included in lines 154-155.
39. Table 3: please provide p-value for the mean score.
p-value included in Table 3.
Section 3.4
40. Please reword lines 215 – 216: Consider: the association between nutritional status and AHEI-2010 score no longer remained significant.
Text was corrected as suggested (lines 235-236).
41. Line 217: do you mean “on average” instead of “in average”?
Text was corrected as suggested (line 237).
Additional comments:
42. Consider moving results about basic sample characteristics to the beginning of the results section (see comment 3).
Done
43. Please make sure that all figures and tables include the survey name and dates of the survey in the title.
We included in all tables and figures titles the survey name (ELANS) and the dates of the fieldwork (2014-2015)
Discussion
44. Please report an overall evaluation of the AHEI-2010 mean score of your sample in the first paragraph of your discussion (i.e. did mean score indicate suboptimal or optimal overall diet quality based on the total score out of 110?). What are implications for suboptimal overall diet quality and health outcomes?
This comment was addressed in the new version of article (lines 251-252).
45. Could you also describe how overall diet quality in your study compares to other studies that have assessed diet quality of Chileans using other diet quality indexes?
Comparisons included in the article (lines 252-259).
46. Lines 229 – 231: If this was something that you examined statistically, it should be written in the Statistical analysis section and reported in the results. Otherwise, please just state that the AHEI-2019 scores were lower in the Chilean sample compared to the U.S. and do not provide a p-value.
We removed the p-value (lines 260 – 265).
47. If you will refer to the United States as US, please write United States (US) in line 229.
As suggested, we fixed this detail in the text.
48. Lines 232 – 234: comment 45 applies.
We removed the p-value.
49. Lines 249 – 250: Even though sodium is the second highest score component, it does not mean that its consumption in Chile is optimal – please do not confuse having the second highest score with meaning the score was high. The mean for sodium was 5.8 out of 10, which is actually not high at all. Unless the AHEI-2010 provides standards for each component (i.e. what scores are considered “high” vs “poor”) please refrain from using this misleading wording. As you state in your discussion, the sodium intake in Chile is suboptimal, and your results support this statement.
We fixed the text as indicated (lines 280-282).
50. Lines 282 – 285: Even though men had a high score in this component (5.2 points) – please reword this to a higher score than women, as a score of 5.2 out of 10 is not really “high” (see comment 47). In addition, consuming 0 drinks/day gives a score of 2.5. Many studies show that consuming no alcohol has health benefits, however, because the AHEI assesses alcohol intake based on moderate consuming, consuming 0 drinks/day has implications for the mean component score. This may be something you want to address in your discussion.
We fixed the text as indicated (lines 322-323). AHEI-2010 defines that nondrinkers have a quarter of the total score because some evidence suggests that moderate consumption have health benefits. Nonetheless, we are aware that there is controversial evidence about this issue but we applied what AHEI-2010 says.
51. Lines 281 – 288: Are you implying that because you only had two 24-hour recalls you may not have captured binge-drinking behavior that has been reported in another Chilean study? I’m not sure you can make this implication based on your results. However, you could state something about limitations of 24-hour recalls in this section. Please rewrite lines 286 -288.
The text was fixed based on this suggestion of the reviewer (lines 377-381 discussing limitations of this study)).
52. Even though the Chilean government provides less stringent fruits and vegetables guidelines, average fruits and vegetables intake in your study was 0.7 servings/day and 2.4 servings/day, respectively. Please elaborate on the potential implications of consuming inadequate fruits and vegetables and health, as you did with other components in previous sections.
The text was fixed based on suggestion of the reviewer (lines 332-333).
53. While the reasoning behind why some of the scores may have scored as the highest or lowest components is interesting, I think this is misleading to readers. Besides trans fat, all of the averages were below 6 out of 10. Sodium and red/processed meat essentially had the same mean score, while the mean value for fruits and SSB were also quite similar. Instead of focusing on the two highest components and the two lowest components, please consider the implications for mean scores as most component scores would likely be considered inadequate. This is why I suggested ordering your table differently and using a radar chart to display results, so you are focused on mean scores rather than order of scores in the table (see comment 32).
Changes were made based on this suggestion.
54. Lines 299 – 301: if the association is not significant, please do not state that individuals with a lower SEL had a higher consumption of legumes.
Text was corrected in lines 339-341.
55. What about a discussion about the red/processed meat component score?
Included in the article (lines 290-296).
56. Please cite this statement (lines 318-319): In addition, we cannot rule out a social desirability bias when participants answered the 24-HR recalls or the beverage frequency questionnaire.
Included in the article (lines 380-381).
57. Could you please write about the clinical significance of the significant associations you found since overall mean AHEI-2010 was <50 across sex, age, SEL, and nutritional status?
Included in the article (lines 360-363).
58. Please remove lines 323 – 324 (We postulate that our lowest AHEI-2010 score group, composed mainly by men, younger subjects and low SEL, may also have an increased risk of CVD and mortality) and make sure your discussion focuses on interpreting your results. This statement is very strong considering your study was cross-sectional and mean AHEI-2010 score was quite low in all terciles.
This statement is based in other cross sectional studies (references 63,64).
59. Please remove lines 329 – 331 (Assuming a causal relationship, a person who increases the AHEI-2010 score in 20 points over a 12-year period may reduce his or her risk of death by nearly 20% in the subsequent 12 years, based on estimations from previous studies). You could instead write “Previous studies suggest that improvements in AHEI-2010 score may reduce risk of mortality over time”.
This statement is based in other cross sectional studies (references 63,64).
60. Please refrain from stating that “this report is one of the best available” (lines 336). Instead, you could state that your findings contribute to a limited body of literature about diet quality in 15 – 65-year-old Chileans living in urban locations.
Text was fixed in lines 382-383.
Conclusions
62. I find that the conclusions stray from the purpose and findings of this cross-sectional study and could be rewritten.
63. This statement on lines 344 – 346 is too strong based on your study design and findings: If so, advances in overall diet quality will lead to noncommunicable chronic disease prevention and increased life quality of our population.
With regard to these two comments, we agree that they go beyond the actual purpose and findings of our study. However, our findings may have implications as indicated in the text. To avoid any misleading conclusions, we have slightly changed that title of this section into Overall Conclusion and Potential Implications (line 386) to more properly represent that actual text.
Additional comments about manuscript:
64. I noticed several minor grammatical errors throughout. Please proofread and make sure tense is consistent throughout manuscript.
Grammatical style was corrected

Round 2
Reviewer 2 Report
I appreciate the thorough response from the authors. All of my comments have been adequately addressed.